# Synthesis of Diesel and Jet Fuel Range Cycloalkanes with Cyclopentanone and Furfural

**Wei Wang [1] 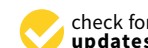, Shaoying Sun [1], Fengan Han [2], Guangyi Li [2], Xianzhao Shao [1] and Ning Li [2,3,*]**

[1] Shaanxi Key Laboratory of Catalysis, School of Chemistry and Environment Science, Shaanxi University of Technology, No. 1 Dong yi huan Road, Hanzhong 723001, China; wangwei@snut.edu.cn (W.W.); sunshaoying1221@163.com (S.S.); xianzhaoshao@snut.eud.cn (X.S.)

[2] CAS Key Laboratory of Science and Technology on Applied Catalysis, Dalian Institute of Chemical Physics, Chinese Academy of Sciences, No. 457 Zhongshan Road, Dalian 116023, China; hanfengan@dicp.ac.cn (F.H.); lgy2010@dicp.ac.cn (G.L.)

[3] Dalian National Laboratory for Clean Energy, No. 457 Zhongshan Road, Dalian 116023, China

* Correspondence: lining@dicp.ac.cn; Tel.: +86-411-84379738

**Abstract:** Diesel and jet fuel range cycloalkanes were obtained in ~84.8% overall carbon yield with cyclopentanone and furfural, which can be produced from hemicellulose. Firstly, 2,5-bis(furan-2-ylmethyl)-cyclopentanone was prepared by the aldol condensation/hydrogenation reaction of cyclopentanone and furfural under solid base and selective hydrogenation catalyst. Over the optimized catalyst (Pd/C-CaO), 98.5% carbon yield of 2,5-bis(furan-2-ylmethyl)-cyclopentanone was acquired at 423 K. Subsequently, the 2,5-bis(furan-2-ylmethyl)-cyclopentanone was further hydrodeoxygenated over the M/H-ZSM-5(Pd, Pt and Ru) catalyst. Overall, 86.1% carbon yield of diesel and jet fuel range cycloalkanes was gained over the Pd/H-ZSM-5 catalyst under solvent-free conditions. The cycloalkane mixture obtained in this work has a high density (0.82 g mL$^{-1}$) and a low freezing point (241.7 K). Therefore, it can be mixed into diesel and jet fuel to increase their volumetric heat values or payloads.

**Keywords:** high density fuel; cyclopentanone; furfural; aldol condensation; hydrodeoxygenation

## 1. Introduction

In recent years, the utilization of renewable biomass resources for the production of fuels [1–6] and chemicals [7–12] has attracted worldwide attention. Lignocellulose is the most abundant and cheapest biomass [13]. Diesel and jet fuel are two very important transport fuels. Pioneered by Dumesic [14,15], Huber [15,16], Corma [17,18] and their collaborators, great efforts were devoted to the synthesis of diesel and jet fuel range hydrocarbons using platform compounds derived from lignocellulose in the past decades [19–26].

Furfural is a bulk chemical which has been obtained on a commercial scale by the hydrolysis and dehydration of hemicellulose [27]. In recent years, series of diesel and jet fuel range chain alkanes were produced by the aldol condensation of furfural and acetone [15,19], methyl isobutyl ketone [28], pentanone [29,30], heptanone [31], levulinic acid [32], and angelica lactone [33], followed by hydrogenation/hydrodeoxygenation or hydrodeoxygenation (HDO). Cyclopentanone can be obtained (at high carbon yields of 62–95.8%) through the aqueous phase selectively hydrogenation and rearrangement of furfural [34–38]. High density fuels were prepared by self aldol condensation of cyclopentanone and hydrodeoxygenation [39–42]. In some studies, diesel and jet fuel range cycloalkanes were produced by the aldol condensation of furfural and cyclopentanone [43–46], followed by HDO. Nevertheless, the aldol condensation products obtained in these reports

(i.e., 2-(2-furylmethylidene)-cyclopentanone and/or 2,5-bis(2-furylmethylidene)-cyclopeantanone) are solid at room temperature. Hence, organic solvents must be used to improve mass transfer efficiency in subsequent HDO reactions. This will cause lower efficiency and higher energy consumption. As a solution to this problem, we developed a new two-step strategy (see Scheme 1). In the first step, 2,5-bis(furan-2-ylmethyl)-cyclopentanone was generated via the aldol condensation/hydrogenation of cyclopentanone and furfural over the metal and solid base catalysts. Due to the saturation of C=C bonds by hydrogenation, 2,5-bis(furan-2-ylmethyl)cyclopentanone is a liquid at room temperature. As a result, it may be directly used in the hydrodeoxygenation procedure without using any solvent. This is favorable in real application. In the second step, the 2,5-bis(furan-2-ylmethyl)-cyclopentanone was converted to diesel and jet fuel range cycloalkanes by the HDO over Pd/H-ZSM-5 catalyst.

## 2. Results and Discussion

### 2.1. Aldol Condensation

First, the aldol condensation of furfural with cyclopentanone was studied over solid base catalysts. According to the analysis of GC (Figure S1 in the Supplementary Materials) and nuclear magnetic resonance (NMR) spectra (Figure S2), the chemical shifts of the target products in $^1$H NMR and $^{13}$C NMR spectra are in good agreement with those reported previously [44]. 2,5-bis(2-furylmethylidene)cyclopentanone was confirmed as the unique product (in Scheme 1). The sequence for 2,5-bis(2-furyl methylidene)-cyclopentanone carbon yields over the solid base catalysts was: CaO > LiAl-HT > MgAl-HT > MgO ≈ CeO$_2$ (see Figure 1). Over the CaO catalyst, 95.4% carbon yield of 2,5-bis(2-furylmethylidene)-cyclopentanone was obtained after reacting for 10 h at 423 K. The yield of 2,5-Bis (2-furylmethylidene)-cyclopentanone on Mg-Al hydrotalcite was 85.7%, while that of 2,5-Bis (2-furylmethylidene)-cyclopentanone on MgO and CeO$_2$ was very low.

**Scheme 1.** Synthetic strategies for the production of C$_{15}$ cycloalkanes with furfural and cyclopentanone [45].

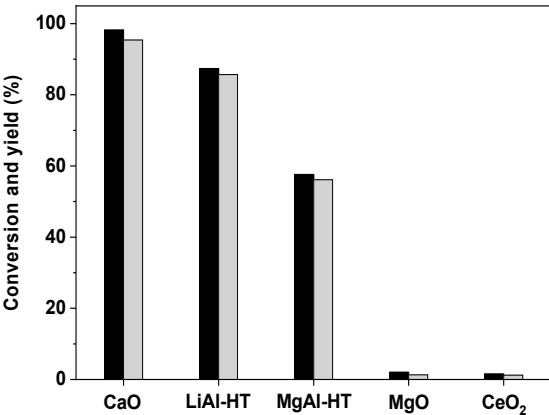

**Figure 1.** Cyclopentanone conversion (black columns) and carbon yield (gray columns) of 2,5-bis(2-furylmethylidene)-cyclopentanone on various solid base catalysts. Experimental conditions: 10 h, 423 K, and 0.84 g cyclopentanone (10 mmol), 1.92 g furfural (20 mmol), and 0.05 g catalyst were used in the tests.

According to the $CO_2$-TPD results (see Figure 2 and Table 1), base site amounts of CaO, LiAl-HT, MgAl-HT, MgO and $CeO_2$ catalysts were 0.16, 0.15, 0.12, 0.04 and 0.02 mmol $g^{-1}$, respectively. The base site amounts of the catalysts were in good agreement with the activity of aldol condensation of cyclopentanone with furfural. The high activity of CaO can be illustrated by its higher alkali strength and higher amount of base sites. Taking into consideration the higher activity, low cost, and good availability of CaO, we consider it as a potential catalyst in future industrial applications.

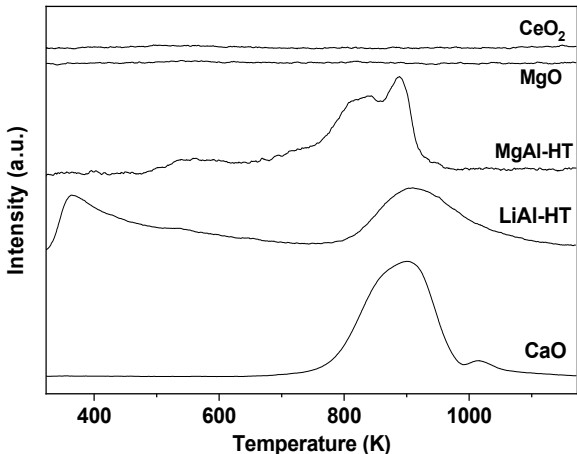

**Figure 2.** $CO_2$-TPD charts of the various solid base catalysts (the specific information of the $CO_2$-TPD experiment is shown in the Supplementary Materials).

**Table 1.** The number of base sites over various solid base catalysts.

| Catalysis | Base Sites Amount (mmol $g^{-1}$) |
|:---:|:---:|
| CaO | 0.16 |
| LiAl-HT | 0.15 |
| MgAl-HT | 0.12 |
| MgO | 0.04 |
| $CeO_2$ | 0.02 |

The influences of catalyst dosage on cyclopentanone conversion and 2,5-bis(2-furylmethylidene) cyclopentanone carbon yield over CaO catalyst were studied as well (see Figure S3a). The results show that the conversion of cyclopentanone raised with the increase of catalyst dosage from 0.01 to 0.07 g. In the meantime, yields of 2, 5-bis(2-furylmethylidene) cyclopentanone increased with the catalyst dosage first, and then leveled off. When the amount of catalyst was 0.05 g, the yield of 2,5-bis(2-furylmethylidene) cyclopentanone was the highest. Therefore, the amount of catalyst in the following part was 0.05 g.

The effects of the reaction temperature on the conversion of cyclopentanone and yield of 2,5-bis(2-furylmethylidene) cyclopentanone were studied. The results are shown in Figure S3b. The conversion of cyclopentanone increased first, and then completely converted as the reaction temperature grew from 373 K to 443 K. At the same time, yield of 2,5-bis(2-furylmethylidene) cyclopentanone grew at first as the reaction temperature ascended, and then remained constant. Consequently, the reaction temperature of 423 K was chosen.

Under the optimum reaction conditions (0.05 g CaO catalyst, 10 h, and 423 K), 98.2% conversion of cyclopentanone and 95.4% carbon yield of 2,5-bis(2-furylmethylidene)-cyclopentanone were achieved.

### 2.2. One Pot Aldol Condensation/Hydrogenation

Subsequently, we developed the one-pot synthesis of 2,5-bis(furan-2-ylmethyl)-cyclopentanone by the aldol condensation/hydrogenation reaction of furfural, cyclopentanone, and hydrogen under the co-catalysis of Pd/C and CaO. Based on GC, NMR and MS spectra analysis (Figures S4–S6),

2,5-bis(furan-2-ylmethyl)-cyclopentanone was identified as the main product. At room temperature, the 2,5-bis(furan-2-ylmethyl)-cyclopentanone exists in liquid state. Consequently, it can be directly used in solvent-free HDO process. According to Scheme 2, 2,5-bis(furan-2-ylmethyl)-cyclopentanone was generated by the hydrogenation of 2,5-bis(2-furylmethylidene)-cyclopentanone from the aldol condensation of furfural and cyclopentanone. To illustrate the advantages of one-pot condensation hydrogenation, we compared the solvent-free, methanol-solvent hydrogenation of 2,5-bis(2-furylmethylidene)-cyclopentanone and one-pot condensation/hydrogenation of furfural and cyclopentanone by Pd/C + CaO. The experimental results are shown in Figure 3. Under the condition of solvent-free Pd/C hydrogenation, conversion of 2,5-bis (2-furylmethylidene)-cyclopentanone and carbon yields of 2,5-bis(furan-2-ylmethyl)-cyclopentanone were 1.1% and 0.8%, respectively.

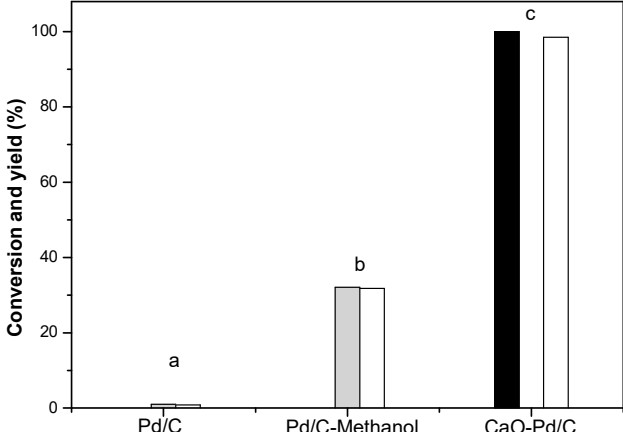

**Scheme 2.** Synthetic strategies for 2,5-bis(furan-2-ylmethyl)-cyclopentanone from 2,5-bis(2-furylmethylde ne)-cyclopentanone hydrogenation [47].

**Figure 3.** Cyclopentanone conversions (black columns), 2,5-bis (2-furylmethylidene)-cyclopentanone conversion (gray columns) and carbon yields of 2,5-bis(furan-2-ylmethyl)-cyclopentanone (white columns) over Pd/C catalysts. Experimental conditions: 10 h, 423 K, and 4.0 MPa $H_2$; (**a**) 2.40 g 2,5-bis(2-furylmethylidene)-cyclopentanone, 0.05 g Pd/C; (**b**) 2.40 g 2,5-bis (2-furylmethylidene)-cyclopentanone, 0.05 g Pd/C and 10.0 mL methanol; and (**c**) 0.84 g cyclopentanone (10 mmol), 1.92 g furfural (20 mmol); 0.05 g CaO and 0.05 g metal catalyst were used in the tests.

When methanol solvent was added to the hydrogenation process, the hydrogenation activity was significantly improved. Conversion of 2,5-bis (2-furylmethylidene)-cyclopentanone and carbon yields of 2,5-bis(furan-2-ylmethyl)-cyclopentanone were 32.1% and 31.8%, respectively. Because methanol can dissolve 2,5-bis (2-furylmethylidene)-cyclopentanone and increase the contact area with hydrogenation active center, the hydrogenation reaction activity of 2,5-bis (2-furylmethylidene)-cyclopentanone was accelerated. Under one-pot condensation/hydrogenation of furfural and cyclopentanone by Pd/C+CaO, cyclopentanone was completely converted and high carbon yield (98.5%) of 2,5-bis(furan-2-ylmethyl)-cyclopentanone was achieved. During one-pot condensation/hydrogenation of furfural and cyclopentanone, furfural and cyclopentanone were not only raw materials, but also as solvents. 2,5-Bis (2-furylmethylidene)-cyclopentanone produced by furfural and cyclopentanone can be dissolved well, which increased the contact area with hydrogenation active center. The reaction rate was accelerated.

Moreover, we studied the activities of CaO + other metal catalysts (PdC, Pt/C, Ru/C, Raney Co and Raney Ni). Based on the results shown in Figure 4, the CaO + Pd/C displayed the best performance among the investigated catalysts. Over it, 100% conversion of cyclopentanone and 98.5% carbon yield of 2,5-bis(furan-2-ylmethyl)-cyclopentanone were obtained, when the reaction was carried out for 10 h at 423 K. Pd/C can selectively hydrogenate carbon–carbon double bonds ($C_1=C_2$ and $C_3=C_4$) of 2,5-bis(2-furylmethylidene)-cyclopentanone, but Pt/C, Ru/C, Raney Co and Raney Ni cannot selectively hydrogenate its carbon-carbon double bonds ($C_1=C_2$ and $C_3=C_4$). The product of carbon-carbon double bonds ($C_1=C_2$ and $C_3=C_4$) of 2,5-bis(2-furylmethylidene)-cyclopentanone hydrogenated is liquid. These products as solvents promoted selective hydrogenation of 2,5-bis(2-furylmethylidene)-cyclopentanone. This can be explained by the high activity of palladium catalyst for high yield of one-pot synthesis of 2,5-bis(furan-2-ylmethyl)-cyclopentanone.

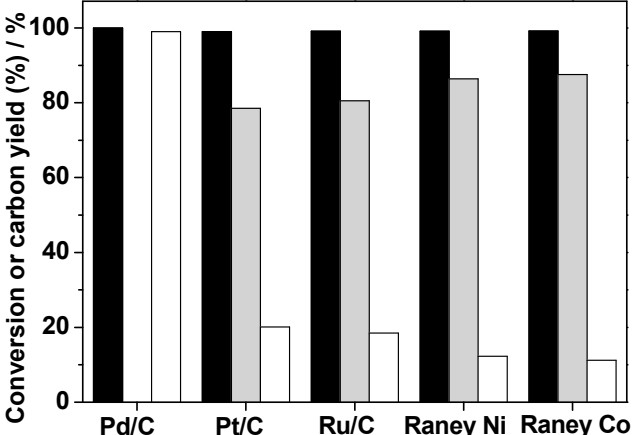

**Figure 4.** Cyclopentanone conversions (black columns) and carbon yields of 2,5-bis(2-furylmethylidene)-cyclopentanone (gray columns), and 2,5-bis(furan-2-ylmethyl)-cyclopentanone (white columns) over CaO and metal catalysts. Experimental conditions: 10 h, 423 K, and 4.0 MPa $H_2$; 0.84 g cyclopentanone (10 mmol), 1.92 g furfural (20 mmol), 0.05 g CaO and 0.05 g metal catalyst were used in the tests.

The influences of catalyst dosage and reaction time on the 2,5-bis(furan-2-ylmethyl)-cyclopentanone carbon yield over the Pd/C-CaO catalyst were investigated (see Figure 5).

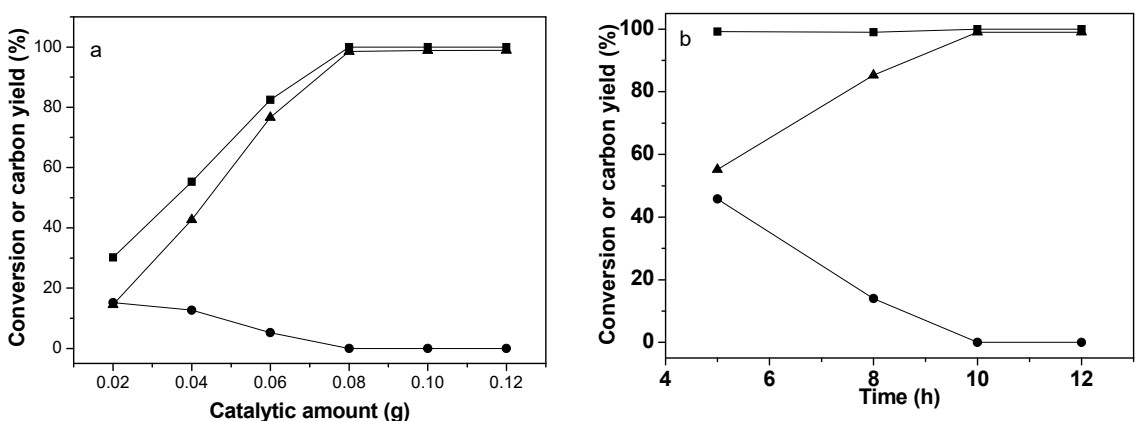

**Figure 5.** Cyclopentanone conversion (■) and carbon yields of 2,5-bis(furan-2-ylmethyl)-cyclopentanone (▲) and 2,5-bis(2-furylmethylidene)-cyclopentanone (●) over the Pd/C-CaO catalyst. Experimental conditions: 423 K, 10 h, and 4.0 MPa $H_2$; 0.84 g cyclopentanone (10 mmol), 1.92 g furfural (20 mmol), 0.08 g Pd/C-CaO catalyst were used in the tests. (**a**) The effect of catalyst dosage. (**b**) The effect of time.

During the change of Pd/C-CaO catalyst from 0.02 g to 0.12 g, the yield of 2,5-bis(furan -2-ylmethyl)-cyclopentanone increased and the yield of 2,5-bis(2-furylmet-hylidene)-cyclopentanone decreased. While the Pd/C-CaO catalyst was 0.08 g, the yield of 2,5-bis(furan-2-ylmethyl)-cyclopentanone reached 98.5%, while cyclopentanone and 2,5-bis(2-furyl methylidene)-cyclopentanone were completely converted. When the reaction time was 10 h, yield of 2,5-bis(furan-2-ylmethyl)-cyclopentanone was highest and 2,5-bis(2-furyl methylidene)-cyclopentanone was completely converted. Under the optimized conditions (423 K, 10 h, and 0.08 g Pd/C-CaO), high carbon yield (98.5%) of 2,5-bis(furan-2-ylmethyl)-cyclopentanone was achieved over the Pd/C-CaO catalyst.

The re-usability of the Pd/C-CaO catalyst was studied as well. To eliminate the disturbance of the residues, the used catalysts were washed thoroughly with tetrahydrofuran (THF) at room temperature after each usage, and then dried for 1 h at 353 K. Based on the results in Figure 6, the Pd/C-CaO catalyst was stable under the investigated reaction conditions.

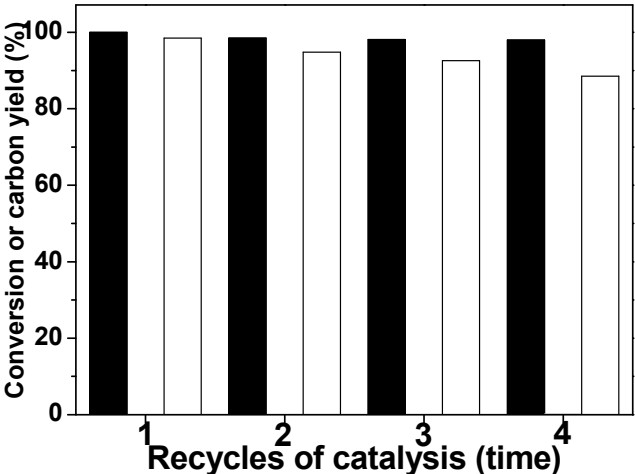

**Figure 6.** Cyclopentanone conversion (black columns) and carbon yields of 2,5-bis(furan-2-ylmethyl)-cyclopentanone (white columns) and 2,5-bis(2-furylmethylidene)-cyclopentanone (gray columns) over the Pd/C-CaO catalyst. Experimental conditions: 423 K, 10 h, and 4.0 MPa $H_2$; 0.84 g cyclopentanone (10 mmol), 1.92 g furfural (20 mmol), 0.08 g Pd/C-CaO were used for the tests.

The 2,5-bis(furan-2-ylmethyl)-cyclopentanone carbon yield varied slightly even after four usages (after the third operation, the slight decrease of activity can be attributed to the loss of catalyst in the process of filtration and drying). Considering the good stability and high activity of the Pd/C-CaO catalyst, we consider Pd/C-CaO catalyst as a potential catalyst for industrial application in the future.

*2.3. Hydrodeoxygenation (HDO)*

The solvent-free HDO of 2,5-bis(furan-2-methyl)-cyclopentanone was investigated over the M/H-ZSM-5 (M = Pt, Pd or Ru) catalysts. Based on GC-MS analysis (Figures S7 and S8), the 2,5-bis(furan-2-ylmethyl)cyclopentanone was totally converted at 573 K, high carbon yields (>66%) of cycloalkanes was obtained over the M/H- ZSM-5 catalysts (see Figure 7). By comparison, the carbon yields (or selectivity) of diesel and jet fuel range cycloalkanes (86.1% and 83.5%) over the 5 wt.% Pd/H-ZSM-5 and 5 wt.% Pt/H-ZSM-5 catalysts were evidently higher than those on the 5 wt.% Ru/H-ZSM-5 catalyst (66.8%). Furthermore, it was also noticed that the carbon yields of C1–C4 light alkanes over the 5 wt.% Ru/H-ZSM-5 was evidently higher than those over the 5 wt.% Pd/H-ZSM-5 and 5 wt.% Pt/H-ZSM-5 catalysts. The lower selectivity of diesel and jet fuel range alkanes over the Ru/H-ZSM-5 catalyst can be explained by the high hydrogenolysis (or methanation) activity of Ru.

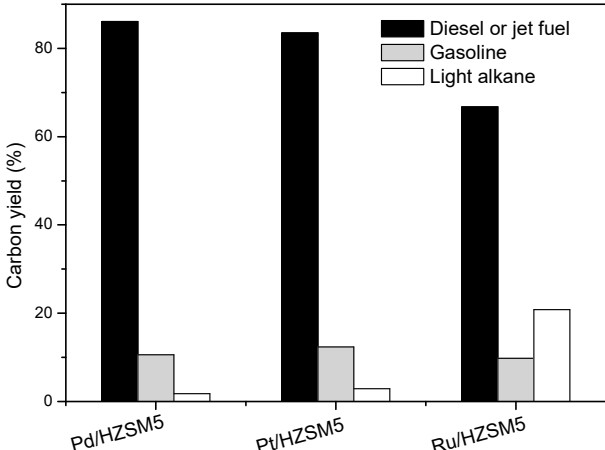

**Figure 7.** Carbon yields of $C_9$–$C_{15}$ diesel alkanes (black columns), $C_5$–$C_8$ gasoline alkanes (grey columns), and $C_1$–$C_4$ alkanes (white columns) over M/H-ZSM-5 (M = Pt, Pd or Ru) catalysts. Experimental conditions: 1.8 g catalyst, 6.0 MPa $H_2$, 573 K, 0.04 mL min$^{-1}$ liquid feedstock flow rate, and 120 mL min$^{-1}$ $H_2$ flow rate.

The stability of Pd/H-ZSM-5 catalyst in the HDO process was studied as well. Based on the results shown in Figure 8, during 24 h of continuous test, the Pd/H-ZSM-5 catalyst was steady under the investigated conditions. According to our determination, the density and the freezing point of the branched cycloalkane mixtures gained in this work were 0.82 g mL$^{-1}$ and 241.7 K, respectively.

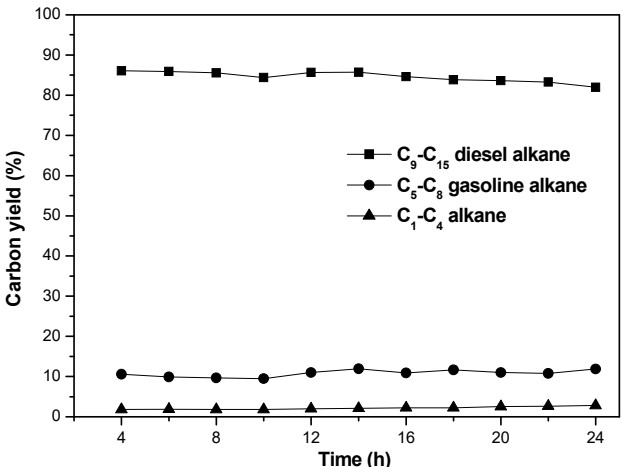

**Figure 8.** Carbon yields of different hydrocarbons over the Pd/H-ZSM-5 catalyst. Experimental conditions: 1.80 g catalyst, 6.0 MPa $H_2$, 573 K, 2,5-bis(furan-2-ylmethyl)-cyclopentanone flow rate of 0.04 mL min$^{-1}$, and hydrogen flow rate of 120 mL min$^{-1}$.

## 3. Materials and Methods

### 3.1. Catalyst Preparation

The CaO, MgO and CeO$_2$ catalysts were purchased from Aladdin Chemical Reagent Co., Ltd. (Shanghai, China). The MgAl hydrotalcite (MgAl-HT) catalyst was homemade according to the method in [15]. To do this, an aqueous solution of magnesium nitrate and aluminum nitrate was added into another aqueous solution of sodium hydroxide and sodium carbonate. This process was conducted at 343 K under vigorous mechanical agitation. After aging at this temperature for 24 h, the as-obtained solid was filtered, repeatedly washed with water until the pH value of the filtrate changed to 7 and dried at 353 K. The dried precursor was calcined for 8 h at 723 K in nitrogen atmosphere. The LiAl hydrotalcite (LiAl-HT) catalyst was synthesized by the method introduced in our previous work.

Typically, an aqueous solution of $Al(NO_3)_3 \cdot 9H_2O$ (125 mL 0.4 mol $L^{-1}$) was slowly added into 300 mL solution of lithium hydroxide (1.5 mol $L^{-1}$) and sodium carbonate (0.08 mol $L^{-1}$). The dried precursor was activated by nitrogen flow for 8 h at 723 K.

The Raney Co and Raney nickel catalysts were obtained from Dalian Tongyong Chemical Co., Ltd. The noble metal catalysts loaded on activated carbon (denoted as M/C, M = Pt, Pd or Ru) were obtained from Aladdin Chemical Reagent Co., Ltd. The metal contents were 5 wt.% in the M/C catalysts on the basis of the information from the supplier. The H-ZSM-5 loaded Pt, Pd, and Ru catalysts were homemade by the incipient wetness impregnation of H-ZSM-5 ($SiO_2/Al_2O_3$ = 25, provided by Nankai University) and the aqueous solutions of $H_2PtCl_6$, $PdCl_2$ and $RuCl_3$, respectively. For comparison, the metal contents of the M/H-ZSM-5 (M = Pt, Pd, and Ru) catalysts were fixed as 5 wt.%.

Pd/C-CaO catalyst was prepared by grinding method. Calcium oxide and palladium–carbon were mixed at a mass ratio of 1:1, and then grinded in a mortar until the mixture was uniform.

### 3.2. Aldol Condensation

The synthesis of 2,5-bis(2-furylmethylidene)-cyclopentanone was realized directly by the aldol condensation of furfural and cyclopentanone in a stainless-steel kettle reactor. In each test, 0.84 g cyclopentanone, 1.92 g furfural and 0.05 g solid base were utilized. The stainless-steel kettle reactor was flushed with nitrogen three times before the test. The mixture of reactant and catalyst was stirred for 10 h at 423 K. Afterwards, the stainless-steel kettle reactor was cooled using ice water. The reaction product was dissolved in 40 mL tetrahydrofuran containing internal standard of cyclohexanone. The liquid products were analyzed by an Agilent 7890A gas chromatograph (GC, Shanghai, China).

### 3.3. One Pot Aldol Condensation

The direct synthesis of 2,5-bis(furan-2-ylmethyl)-cyclopentanone was conducted by the one-pot cascade reaction of cyclopentanone, furfural, and hydrogen using a 20 mL stainless-steel kettle reactor (Internal diameter: 2.0 cm, high: 8.0 cm) with hydrogen pressure gauge. In each test, 0.84 g cyclopentanone, 1.92 g furfural, 0.05 g solid base and 0.05 g Raney metal (or M/C) catalyst were utilized. The reactor was flushed by hydrogen three times before the test. Subsequently, hydrogen was added into the reactor to make the system pressure up to 4.0 MPa. The mixture of reactant and catalyst was stirred for 10 h at 423 K, and then quenched to room temperature. The liquid product was isolated with catalyst by centrifuge and analyzed by the Agilent 7890A gas chromatograph.

### 3.4. Hydrodeoxygenation

The hydrodeoxygenation of 2,5-bis(furan-2-methyl)-cyclopentanone was implemented at 573 K under solvent-free condition. In each test, 1.8 g Pd-HZSM-5 catalyst were used. This catalyst was reduced in-situ in a stainless-steel tube reactor ($\phi$6.0 mm × 1.5 mm, length: 350 mm) for 2 h at 773 K by hydrogen flow (120 ml $min^{-1}$). Then, the reactor temperature dropped to 573 K and stabilized at this value for 0.5 h. 2,5-Bis (furan-2-methyl)-cyclopentanone was transported to the reactor by high pressure pump (0.04 mL $min^{-1}$) with hydrogen (120 mL $min^{-1}$). Mixture products were collected in the gas–liquid separation tank and back pressure regulator (maintaining system pressure at 6.0 MPa). Gas phase samples were analyzed on-line by Agilent 6890N GC. The liquid samples were collected regularly from the bottom of the separation tank and diluted and analyzed by Agilent 7890A gas chromatograph.

## 4. Conclusions

Diesel and jet fuel range cycloalkanes were synthesized from furfural and cyclopentanone by the one-pot aldol condensation/hydrogenation, and hydrodeoxygenation (HDO) in solvent-free conditions. Among the investigated catalysts, Pd/C-CaO displayed the highest activity for aldol condensation/hydrogenation reaction. Over it, high carbon yield (98.5%) of 2,5-bis(furan-2-ylmethyl)-cyclopentanone was acquired under mild conditions (4.0 MPa $H_2$, and 423 K).

2,5-bis(furan-2-ylmethyl)-cyclopentanone exists in liquid state at room temperature. As a result, it can be directly used for the solvent-free hydrodeoxygenation process. This is advantageous in industrial application. Pd/H-ZSM-5 was discovered to be a highly active and stable catalyst for the HDO of 2,5-bis(furan-2-ylmethyl)-cyclopentanone. Over it, 86.1% carbon yield of diesel and jet fuel range cycloalkanes was reached. No significant inactivation was noticed during the 24 h time on stream. According to our measurements, the cycloalkane mixture as obtained has a freezing point of 241.7 K and a density of 0.82 g mL$^{-1}$. As a potential application, it can be mixed with diesel and jet fuel to enhance their volume calorific values or payloads.

**Supplementary Materials:** The following are available online at http://www.mdpi.com/2073-4344/9/11/886/s1, Figure S1: GC chromatogram of the aldol condensation product of cyclopentanone and furfural, Figure S2: $^1$H and $^{13}$C NMR spectra of 2,5-bis (2-furylmethylidene) cyclopentanone, Figure S3: The influences of catalyst dosage and temperature on cyclopentanone conversion, and 2,5-bis(2-furylmethylidene)cyclopentanone conversion carbon yield, Figure S4: GC chromatogram of the 2,5-bis(furan-2-ylmethyl)cyclopentanone from the one-pot aldol condensation/hydrogenation reaction, Figure S5: $^1$H and $^{13}$C NMR spectra of 2,5-bis(furan-2-ylmethyl) cyclopentanone, Figure S6: Mass spectrogram of the 2,5-bis(furan-2-ylmethyl)cyclopentanone, Figure S7: GC chromatogram of the products from the solvent-free HDO of 2,5-bis(furan-2-ylmethyl)-cyclopentanone, Figure S8: Mass spectrogram of the 1,3-dipentylcyclopentane.

**Author Contributions:** Data curation, W.W. and S.S.; Formal analysis, F.H.; Investigation, G.L. and X.S.; Writing—review & editing, N.L.

**Funding:** This research was funded by the National Natural Science Foundation of China (Nos. 21776273, 21721004, and 21690082), DNL Cooperation Fund, CAS (DNL180301), the Strategic Priority Research Program of the Chinese Academy of Sciences (XDB17020100), the National Key Projects for Fundamental Research and Development of China (2016YFA0202801), and Dalian Science Foundation for Distinguished Young Scholars (No. 2015R005). Dr. Wei Wang appreciates Natural Science Basic Research Plan in Shaanxi Province of China (2018JM2048), the Open Project of Shaanxi Key Laboratory of Catalysis (No. SLGPT2019KF01-24), and the Funds of Research Programs of Shaanxi University of Technology (No. SLGQD13(2)-1) for financial support.

**Acknowledgments:** This work was supported by Natural Science Basic Research Plan in Shaanxi Province of China (2018JM2048) and the Open Project of Shaanxi Key Laboratory of Catalysis (No. SLGPT2019KF01-24).

**Conflicts of Interest:** The authors declare no conflict of interest.

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
