# Peer review of "Synthesis of Diesel and Jet Fuel Range Cycloalkanes with Cyclopentanone and Furfural"

_catalysts, doi:10.3390/catal9110886_

Round 1

Reviewer 1 Report

My comments are detailed in the review.

Author Response

First of all, the authors would like to thank reviewer for carefully reading our manuscript and giving us some constructive suggestions. Experimental study on Synthesis of diesel and jet fuel range cycloalkanes with cyclopentanone and furfural. The manuscript is a valuable contribution to the area, a topic which occupies and will occupy generations of researchers [1]. The overall impression of the manuscript is good. It is easily readable. However, this manuscript needs a minor revision prior to publication. For example, a last grammar and vocabulary check by a native English speaker would be appropriate. Answer: Thanks a lot for your positive comments. Following your suggestions, we made some modifications to the manuscript and the supporting information. Grammar and vocabulary of this revised manuscript was check by a native English speaker. The high duplication content has also been modified. We hope that you can be satisfied with the updated version. The layout of the document is a first surprise: the materials and methods chapter (§ 3) is placed after the results chapter (§2). What are the reasons of a such choose? I suggest you put the materials and methods chapter as §2 like the majority of the authors and so, avoid misunderstanding. Answer: We prepared the manuscript according to the template of the Catalysts journal. In this template, the materials and methods chapter (§ 3) is placed after the results chapter (§2). Therefore, this is something we can not change. Sorry for this. Concerning the materials and methods chapter, the precise reference of the used catalyst is not given. I suggest you give it to ensure the reproducibility of your tests. Answer: The CaO, MgO and CeO2 catalysts were bought from Aladdin Chemical Reagent Co., Ltd. The MgAl hydrotalcite (MgAl-HT) catalyst was homemade according to literature [15]. Moreover, CeO2 is mainly used in nanostructured forms. As a result, could you precise it? Moreover, it would be interesting to compare that point through your studies to identify a cause of your different reactivities. Answer: Cerium dioxide is purchased from commercial products, not nanostructured forms. In case of nanomaterial utilization in your set up, I would like to inform you that recent works on nanoparticles showcase potential risks of nanoparticle releases and allow a more balanced benefit/risk analysis. For example, many studies highlight nanoparticle emissions due their presences. Cases of nanoparticle exposure in the field of occupational hygiene at workplaces have been reported [2]. Answer: In this work, we focused on calcium oxide, which is obtained as a commercial product, not in the nanostructured forms. As we know, the CaO is not so harmful. When it exposed to air for long time, it will react with CO2 and form CaCO3. Therefore, the potential risks of nanoparticle releases can be ignored. Secondly, references are missing on that point. You should cut-and-paste related references below and put them in your manuscript. Moreover, some references from this journal can be added. Answer: References [37-39] of this journal have been added. I will now share my observations in order of appearance in the manuscript. Title: The title is well chosen and indicates clearly the scope and topic of your contribution. Graphical abstract: “A Graphical Abstract is a single, concise, pictorial and visual summary of the main findings of the article. This could either be the concluding figure from the article or a figure that is specially designed for the purpose, which captures the content of the article for readers at a single glance. Please see examples below.” I this perspective, your graphical abstract could be improved and become more ‘catchy’ and self-explaining. Answer: We have improved the Graphical Abstract as you suggested. Highlights Research highlights ‘are a short collection of bullet points that convey the core findings and provide readers with a quick textual overview of the article.’ The proposed highlights are a good choice in this view, however you might want to check whether these are ‘stand-alone’ choices, i.e. if the reader could understand the scopes and respective findings. I think there is room for improvement, even if the number limitation of characters is obviously a constraint. Check the use of each word, Answer: This manuscript is written in strictly accordance with the writing requirements and templates of Catalysts journals. There is no highlight subject in the manuscript. Abstract: ‘An abstract is a brief summary of a research article, thesis, review, conference proceeding, or any in-depth analysis of a particular subject and is often used to help the reader quickly ascertain the paper's purpose.’ In this view, your abstract could be improved, similarly to the previous point, - By more clearly indicating the purpose of your research. The need for research in this field is certainly a strength of your paper, so tell people about. - You might give a hint to the quality of your results, i.e. relative improvement, repeatabilities etc. The abstract remains vague with respect to this. Answer: The abstract has been modified appropriately, and the modified content is displayed in blue. Introduction: ‘The introduction leads the reader from a general research issue or problem to your specific area of research. It puts your research question in context by explaining the significance of the research being conducted. This is usually done by summarizing current understanding (research to date) and background information about the topic. This is followed by a statement of the purpose of your research issue or problem. This is sometimes followed by a hypothesis or a set of questions you attempt to answer in your research. You may also explain your methodology (how you will research this issue) and explain what your study can reveal. It also may contain a summary of the structure of the rest of the paper.’ The introduction as a whole is corresponding to these needs and very well written. § Results Line 54: You write: ”First of all, the aldol condensation of furfural with cyclopentanone was studied over solid base catalysts. According to the analysis of GC (Figure S1 in the supplementary materials) and nuclear magnetic resonance (NMR) spectra (Figure S2), the chemical shifts of the target products in 1H NMR and 13C NMR spectra are good agreement with those reported earlier (refence)”. I suggest the modifications in the sentence following my suggestions (red mark). Answer: We have modified the manuscript according to your suggestion. Line 65 scheme 1: Moreover, a reference explaining the chemical reactions (with the catalysts you use) of aldol condensation would be an improvement Answer: Scheme 1: Moreover, a reference [45] was introduced to explain the chemical reaction of aldol condensation. Line 66 : The figure 1 is not clear explain the grey / black meaning. Answer: We have explained the meaning of grey column and black column in line 70 at the page 2 of the revised manuscript. Line 112: a reference explaining the chemical reactions (with the catalysts you use) of hydrodeoxygenation would be appreciated. Answer: In line 117 at page 4 of the revised manuscript, a reference [47] was introduced to explain the chemical reactions (with the catalysts you use) of hydrogenation. materials and methods chapter Generally, the Methods chapter should allow for getting all the basic details so that the experiments can be reproduced, mentioning all the appropriate controls, including appropriate citations included and the reference of each single material used. I think this chapter in this manuscript fulfills this purpose, but, however, lacks of sufficient basic information of this type: Please add dimensions, parameter ranges etc. In view of its reproducibility, the efforts allowing for improving this might be indicated (see previous remarks on that chapter). Conclusions: The conclusions chapter reminds well the reader of the strengths of your central points and summarizes the evidence supporting these. Answer: Following your suggestion, we have measured the dimensions of the reactors. The information has been given at the page 8 of the revised manuscript. References suggested (MDPI style devoted to catalysts) Answer: References were modified according to MDPI style devoted to catalysts.

Reviewer 2 Report

This paper, although not tremendously innovative, reports some research on the comparison of activities between some catalytic systems for the aldol condensation / hydrogenation reaction. The 2,5-bis (furan-2-methyl)-cyclopentanone obtained, which is in the liquid state an RT, allows to perform the hydrodeoxygenation process without solvent. This fact represents an indisputable advantage for the industrial production of diesel and jet fuels with the use of renewable biomass resources.

the following point is brought to the attention of the authors:

- reference 41 is mentioned in the text, but is not reported.

- In the text the authors refer to the figures shown in the supplementary material, without however commenting on the results. The work should be understood without the use of additional material, which should only be used to investigate further. See e.g. pag. 3: “The effect of the reaction temperature on the conversion of cyclopentanone and yield of 89 2,5-bis(2-furylmethylidene) cyclopentanone were studied and the results were shown in Figure S 3b.”

Author Response

First of all, the authors would like to thank reviewer for carefully reading our manuscript and giving us a very constructive suggestion.

This paper, although not tremendously innovative, reports some research on the comparison of activities between some catalytic systems for the aldol condensation / hydrogenation reaction. The 2,5-bis (furan-2-methyl)-cyclopentanone obtained, which is in the liquid state an RT, allows to perform the hydrodeoxygenation process without solvent. This fact represents an indisputable advantage for the industrial production of diesel and jet fuels with the use of renewable biomass resources.

Answer: Thanks a lot for your positive comments. Following your suggestion, we made some modifications to the manuscript. We hope that you can be satisfied with the updated version.

the following point is brought to the attention of the authors:

- reference 41 is mentioned in the text, but is not reported.

Answer: Thanks for reminding. We have replaced reference 41 with reference [15] in line 210 at page 7 of the revised manuscript.

- In the text the authors refer to the figures shown in the supplementary material, without however commenting on the results. The work should be understood without the use of additional material, which should only be used to investigate further. See e.g. pag. 3: “The effect of the reaction temperature on the conversion of cyclopentanone and yield of 2,5-bis(2-furylmethylidene) cyclopentanone were studied and the results were shown in Figure S 3b.”

Answer: We have added the following comments in line 94 at the page 3 of the revised manuscript.

The conversion of cyclopentanone increased first, and then completely converted as the reaction temperature increased from 373 K to 443 K. Moreover, the yield of 2,5-bis(2-furylmethylidene) cyclopentanone increased at first with the reaction temperature, and then constanted. Therefore, the chosen reaction temperature was 423 K.

Round 2

Reviewer 2 Report

In the revised version of the paper the authors took into account what was underlined in the review of the original manuscript.
So, in my opinion, the document can be accepted for publication on Catalyst.